# Exploring unintended pregnancy journeys among women with psychiatric vulnerability using interpretative phenomenological analysis

Noralie N. Schonewille[1,2,3], Elena Soldati 🆔[1,2,3]*, Monique J.M. van den Eijnden[1,4], Nini H. Jonkman[5], Maria G. van Pampus[6], Thomas Zoon 🆔[1], Odile A. van den Heuvel[2,7,8], Birit F.P. Broekman[1,2,3]

**1** OLVG (Department of Psychiatry and Medical Psychology), Amsterdam, The Netherlands, **2** Amsterdam UMC, Vrije Universiteit Amsterdam (Department of Psychiatry), Amsterdam, The Netherlands, **3** Amsterdam Public Health, Mental Health Program, Amsterdam, The Netherlands, **4** MIND, (Team Knowledge, Innovation and Research), Amersfoort, The Netherlands, **5** OLVG, (Department of Research and Epidemiology), Amsterdam, The Netherlands, **6** OLVG, (Department of Gynecology and Obstetrics), Amsterdam, The Netherlands, **7** Amsterdam UMC, Vrije Universiteit Amsterdam (Department of Anatomy & Neuroscience), Amsterdam, The Netherlands, **8** Amsterdam Neuroscience, Compulsivity, Impulsivity & Attention program, Amsterdam, The Netherlands

* e.soldati@olvg.nl

## Abstract

### Background

It is known that women with unintended pregnancies (UPs) experience many challenges. Women with psychiatric vulnerability may face specific concerns regarding the transmission of psychiatric vulnerability, parenting skills and bonding capacities with their offspring. This study aimed to explore how women with psychiatric vulnerability experience UPs.

### Methods

This is a prospective qualitative study using semi-structured interviews during pregnancy and after delivery regarding the experiences of women with UPs and psychiatric vulnerability and involved partners. Follow-up interviews were conducted three to six months after delivery. Interpretative phenomenological analysis was employed to analyze the data.

### Results

Women with psychiatric vulnerabilities described unintended pregnancies as complex events, often marked by ambivalent pregnancy intentions, concerns about generational trauma, and fears about parental adequacy. The pregnancies triggered heightened psychiatric symptoms, resurfacing childhood memories, and concerns about stigma, yet also motivated participants to seek support from mental health

**Data availability statement:** The metadata for this study have been made publicly available to facilitate transparency and initial exploration (https://doi.org/10.34894/LISNEO). Access to the full database, which contains detailed information and study results, is restricted and will breach the confidentiality requirements approved by our institutional research ethics board, and requires explicit permission, due to the qualitative nature of the data. The database contains confidential, personal, and sensitive information that could compromise participant privacy, particularly given the inclusion of individuals from vulnerable communities. In accordance with the consent form, only disguised extracts may be used in scientific publications. Researchers interested in accessing the full database must contact the local review committee at OLVG (ACWO) via email (acwo@olvg.nl), which, in accordance with current laws and regulations, will review requests to ensure data privacy and ethical standards are maintained. The ACWO operates under the supervision of the Board of Directors of OLVG.

**Funding:** This research was funded by ZonMw, grant number 554002007.

**Competing interests:** The authors have declared that no competing interests exist.

**Abbreviations:** IPA, interpretative phenomenological analysis; MHP, mental health professional; UP, unintended pregnancy; GA, gestational age; PTSD, posttraumatic stress disorder; OCD, obsessive compulsive disorder; OCPD, obsessive-compulsive personality disorder; LAT, living apart together.

professionals and trusted others. Women adopted coping strategies such as focusing on the future, seeking distraction, and accepting support to manage emotional challenges. Across pregnancy and postpartum, many participants reported developing strong prenatal and postnatal bonding with the newborn, creating safety nets, and making intentional behavioral changes to support their babies. For several women, the unintended pregnancy ultimately fostered personal growth and contributed to an improvement in mental well-being.

## Conclusions

This study elucidates the experiences of unintended pregnancies in women with psychiatric vulnerability. Our findings show that for women with psychiatric vulnerability, UPs may also become a window of opportunity for treatment, personal growth and create a safety net for the baby and oneself. This work may help mental healthcare providers to support comprehensively expectant parents who decide to continue UPs.

## Background

Unintended pregnancies (UPs) (pregnancies that are mistimed and/or unwanted) account for up to 50% of all pregnancies worldwide [1]. UPs have a tremendous impact on pregnant women, their newborns and society, as they are associated with a higher risk for perinatal depression and lower birth weight, impaired parent–child interactions, increased parenting stress, and a substantial social and financial burden [2–7]. This impact is especially relevant to patients with psychiatric vulnerability (present or past psychiatric disorders), as they have an increased risk for UPs and an increased risk for adverse pregnancy and birth outcomes irrespective of pregnancy planning status [8–13]. Previous literature showed that psychiatric vulnerability could contribute to challenges with family planning. Family planning, according to the WHO, consists in practices that "allow people to attain their desired number of children, if any, and to determine the spacing of their pregnancies" [14]. Patients with psychiatric vulnerability can experience difficulties in adhering to contraceptives, planning and reproductive autonomy [15–19]. For example, women with schizophrenia or bipolar disorders may suffer from cognitive and behavioral alterations that affect their reproductive planning and decisions. Studies found an association between depressive symptoms, or depressive symptoms co-occurring with stress, and inconsistent contraceptive use [15–19]. Stress and depressive symptoms may influence processes such as decision-making, risk assessment or increased side effects of contraceptive and therefore cause discontinuation [19]. Finally, one study found that women pertaining to a marginalized group and with a low socioeconomic status were more at risk for developing depression symptoms as well as UPs than women of majority groups or privileged groups [20].

However, these findings are based on quantitative studies, which do not explore the individual experiences among pregnant women and their partners. In contrast, some qualitative studies have explored the experiences of UPs of women without

psychiatric vulnerability. These studies describe pregnancy intentions as complex processes influenced by an interplay of both extrinsic factors, such as social, financial and relationship status, as well as intrinsic factors, such as the emotional and mental health status of the pregnant woman [21,22]. The impact of UPs on the lives of the participants included experiences of guilt, self-blame, stress, worry, interpersonal conflict and societal stigma [23,24]. Previous research on family planning perceptions of pregnant women with psychiatric vulnerability revealed specific concerns such as the transmission of psychiatric vulnerability, and anticipated problems with parenting skills and bonding capacities with children [25–30]. Partners' perspectives are often overlooked in research on UPs, even though they may play an important role in the decision-making process [31].

To date, the experiences of women with psychiatric vulnerability faced with UPs have not been described.

This study aims to explore how women with psychiatric vulnerability experience unintended pregnancies. The objectives are to (1) identify and describe how UP occurs among women with psychiatric vulnerability and (2) factors that influence decisions around maintaining the6 pregnancy. Other objectives concern the mother-child bonding during pregnancy and the transition to parenthood. When possible, partners of the participants will also be interviewed.

To support women with psychiatric vulnerability and UPs emotionally and practically, it is crucial to understand the challenges these women encounter and their needs in order to develop family planning programs tailored to the needs of women with psychiatric vulnerability.

## Methods

### Research Design Overview

A prospective qualitative research methodology was adopted to gain a deep understanding of women's lived experiences over time [32]. A prospective study can best answer our research questions as participants can be interviewed during their pregnancy and after birth and therefore their experiences can be analyzed as they unfold [32].

We conducted semi-structured interviews during and after pregnancy (see Appendix 1). The analysis was grounded in the phenomenological tradition, and we adhered to the principles of interpretative phenomenological analysis (IPA) [33,34]. The IPA fits the aims of the current study, as it sets priority to the individual experience, aims at providing a detailed and nuanced analysis of people's experiences, is appropriate for studying life events [33,35] and focuses on a specific context [36]. Additionally, IPA appears suitable for amplifying the concerns of underrepresented groups, such as women with psychiatric vulnerability whose experiences are currently understudied [37]. The consolidated criteria for reporting qualitative research (COREQ) checklist was followed to assure transparency [38].

### Researcher description

The position of the researchers holds relevance in any study conducting IPA, as it adheres to the double hermeneutic position: the researchers attribute meaning to how the participants attribute meaning to their own experience [39]. NS is a medical doctor with experience in the field of obstetrics and perinatal psychiatry; HS is a midwife with special knowledge of and interest in international public health; ES is a medical doctor with experience in psychiatry and obstetric care for marginalized groups of pregnant women; ME is a postdoc researcher who has lived experience in perinatal psychiatry and is an employee of organization MIND (the Dutch mental health patient and family umbrella organization); NJ is a postdoc researcher with methodological expertise; MP is a senior researcher and perinatologist; TZ is a researcher and hospital psychiatrist; OH is a senior researcher, psychiatrist with expertise in hospital psychiatry and has lived experience with UPs and psychiatric vulnerability; and BB is a senior researcher and psychiatrist with perinatal expertise. As a research team, we also acknowledge that our identities and social positions may have influenced the research process. Most members of the team identify as women and have a clinical professional background which may have shaped our empathy toward pregnant participants as well as our assumptions about clinical care. Several researchers are parents themselves or have

personal experience with psychiatric vulnerability or unintended pregnancy, which may have increased our sensitivity to participants' accounts, but also required ongoing reflexivity to avoid over-identification. Despite diversity in cultural and socioeconomic backgrounds within the research team, all researchers occupy positions of relative educational and institutional privilege. This shared positionality may have contributed to power asymmetries in interactions with participants. Furthermore, two researchers (NS and ES) were junior researchers and PhD students, thus dependent on the more senior colleagues. We discussed these dynamics throughout data collection and analysis, and reflexively considered how our identities, professional roles, and lived experiences influenced the questions we asked, our interpretation of participants' narratives, and the meaning-making process.

## Participant selection and recruitment

Recruitment took place between March the 1st 2022 and February the 28th 2023 among all women who visited the psychiatric perinatal outpatient clinic. We included women who perceived their pregnancies as unintended, were aged 18 or older and were proficient in Dutch or English. We excluded women who could not provide informed consent due to intellectual disabilities or had a florid psychotic state, as evaluated by a consulting psychiatrist. Partners of pregnant women were invited for the study. Eligible women were informed about the research during routine clinic visits between nine and 34 weeks of gestation and were invited by a researcher via email or telephone. The consulting psychiatrist did not participate in the interviews. When pregnant women agreed to participate, patient information documents and informed consent forms were sent via email. Each pregnant woman was asked whether a partner was involved and if this partner wanted to be interviewed too, separately or together. When the partner agreed to participate, they also signed an informed consent form. Participants and partners were invited for two consecutive interviews—one during pregnancy and one within two to six months postpartum, depending on availability of the participant. The invites were sent six weeks after birth. Participants received reimbursement for their time and travel expenses. Purposeful sampling enabled the inclusion of participants with UPs, a willingness to discuss the topic and varying psychiatric vulnerability. Moreover, snowball sampling was used to recruit the partners of the participants. We followed the IPA principle of including a maximum of 10–15 participants in order to support the detailed analysis of each individual participant [34,40,41]. Finally, participants were offered to bring a support person (other than the partner) to the interviews, as the interview topic might be perceived as sensible.

## Data collection

The interview guide was developed based on qualitative data collected from focus groups with the MIND mental health panel (see Appendix 1) [42]. Throughout the study, the interview guide was modified to accommodate any new themes that emerged during the interviews. All interviews were conducted by two female researchers(NS and ME): one as an interviewer and one as an observer (rotating roles). The interviews were conducted either via videoconference using Zoom or at the hospital, depending on the participant's preference, and were audio recorded. Interviews were recorded by a voice recorder (not connected to internet, Bluetooth or Wifi). The voice recorder was started after an informal introduction of both interviewee, interviewer, and observer. After the interview had taken place, the recordings were uploaded to a secured hospital-based data management system. During the interviews, the researchers made notes and captured nuances such as body language and facial expressions to supplement the audio recordings for a more comprehensive analysis.

## Data Analysis

Demographic data were described per individual patient and analyzed descriptively, using mean or number for group summaries. The recorded interviews were manually anonymized and transcribed verbatim by the researchers (NS and ES) independently. The transcripts were checked a second time (NS and ES) for possible errors. We generated

summaries and shared them with participants for credibility checks. We generated summaries and shared them with participants for credibility checks. Data analysis was conducted by two researchers (NS, ES) and a patient investigator (ME), who offered diverse perspectives to enrich trustworthiness. As the researchers speak both Dutch and English, no translation was necessary and the analysis was performed in the same language as the interview was performed. The selected quote for the manuscript were translated by the researchers. The IPA was conducted using ATLAS.ti V9.1. We adhered to the steps described by Charlick et al. and applied an inductive approach to the data analysis: reading and rereading transcripts (NS, ES, ME), initial noting of transcripts, by making descriptive and conceptual comments, including emotional expressions (memos are consulted in this step) (NS, ES), developing emergent themes (NS, ES), searching for connections across emergent themes (NS, ES), moving to the next participant (NS, ES), and looking for patterns across participants (NS, ES) [33]. In this latter step, an analytical framework was built, including participants' narratives as columns and emergent themes as rows. Thus, it was possible to initially focus on the individual narratives before making interpretations at the group level. The last step included examining the analysis in a group meeting to reduce possible personal biases and enhance transparency (all authors). Credibility was therefore enhanced through investigator triangulation [43].

### Ethics approval

This study was performed in accordance with the principles of the Declaration of Helsinki. The Medical Ethics Research Committee of the United Nieuwegein declared the study as not part of the Human Subjects Research Law, protocol code W21.304 and date of declaration 07-01-2022. Local approval was granted by the accredited Ethics Committee on 28-04-2022.

## Results

### Demographics

Eleven participants were interviewed during pregnancy: nine pregnant women and two male partners (one interviewed together with his partner and one interviewed separately from his partner). Of the eleven participants, six also participated in the postpartum interview (five women and one male partner). The attrition rate was 45.45%. The variation in timing for the postpartum interview depended on the availability of the participant. The loss to follow-up of five interviewees was due to not having mind space to participate (n = 1), not responding to the invitation (n = 3), and loss of pregnancy (n = 1). None of the participants requested alterations to their narrative in the summary. Table 1 shows the demographic and psychosocial characteristics of the participants. The mean age of the participants was 32.3 years, of the partners was 29.5 years, most of them were primipara except for one. Five women were born in the Netherlands, four in other countries (North America, Europe and South America) and the two partners in the Netherlands. The average duration of the antepartum interviews was 45 minutes and of the postpartum interviews was 48.6 minutes.

Four main themes were identified (see Fig 1). The order of the themes displays the chronological approach of the two consecutive interviews during and after pregnancy.

### Ascribing meaning to an unintended pregnancy

**Conception by misconception.** Most participants actively tried to avoid pregnancy by using contraception, such as copper intrauterine device, a birth control app or condoms. In these instances the discovery of the pregnancy was marked by astonishment.

*"It's not like we had unsafe sex, we just used condoms in the meantime, but still, somehow, I got pregnant. We were on vacation in [country], and I was supposed to have my period that week, and I was a bit bloated, and then I thought; well, how is that possible?"* (Participant 1)

 

**Table 1. Demographics of participants.**

| Participant | Age | Parity | Employed | Partner status | Self-reported psychiatric vulnerability | Self-reported childhood adverse experience(s) | Discovery pregnancy in trimester | First interview (GA) | Postpartum interview (weeks postpartum) |
|---|---|---|---|---|---|---|---|---|---|
| 1 | 29 | 0 | Yes | Living in | Bipolar disorder type I | Yes | 1st | 36 weeks | 10 weeks |
| 2 | 32 | 0 | Yes | Living in | PTSD anxiety disorder | Yes | 1st | 34 weeks | 8 weeks |
| 3 | 39 | 0 | Yes | Living in | Borderline personality disorder Depressive disorder | Yes | 1st | 24 weeks | – |
| 4 | 33 | 0 | Yes | Married | Depressive disorder Anxiety disorder | Yes | 1st | 35 weeks | 16 weeks |
| 5 | 39 | 0 | Yes | Married | Anxiety disorder Panic disorder | Yes | 1st | 9 weeks | – |
| 6 | 35 | 0 | Yes | LAT | OCD | Yes | 1st | 27 weeks | 24 weeks |
| 7 | 24 | 0 | Yes | Living in | Attention Deficit Hyperactivity Disorder | Yes | 1st | 23 weeks | 14 weeks |
| 8 | 28 | 2 | Yes | Married | Anxiety disorder Depressive disorder | Yes | 1st | 13 weeks | – |
| 9 | 32 | 0 | Yes | Single | OCD, OCPD, PTSD | Yes | 1st | 20 weeks | – |
| Partner of participant 1 | 31 | 0 | Yes | Living in | No self-reported psychiatric vulnerability | Yes | 1st | 38 weeks | – |
| Partner of participant 2 | 28 | 0 | Yes | Living in | No self-reported psychiatric vulnerability | No | 1st | 34 weeks | 8 weeks |

* GA, gestational age; PTSD, posttraumatic stress disorder; OCD, obsessive compulsive disorder; OCPD, obsessive compulsive personality disorder; LAT, living apart together.

* GA, gestational age; PTSD, posttraumatic stress disorder; OCD, obsessive compulsive disorder; OCPD, obsessive compulsive personality disorder; LAT, living apart together.

Other participants had misconceptions about their own fertility, due to older age or eating disorder and therefore did not use contraceptives. Participant 2 further commented that her pregnancy also caused relief: now that she had conceived, she was positively fertile.

"[…] with that eating disorder, I didn't know because I got my period so late, whether I was even fertile." (Participant 2)

Beside participant 2, no other participant indicated that her psychiatric vulnerability was related to the UP.

**Not sure of my intentions.** The pregnancy intentions of several participants were ambivalent. This was illustrated by contradictory behavior toward preventing a pregnancy. For example participant 2 used folic acid but at the same time did not wish to become pregnant. The fear of consciously considering motherhood contributed to this ambivalence; participants 2, 3 and 7 were happy that the UP itself had prevented them from making the decision.

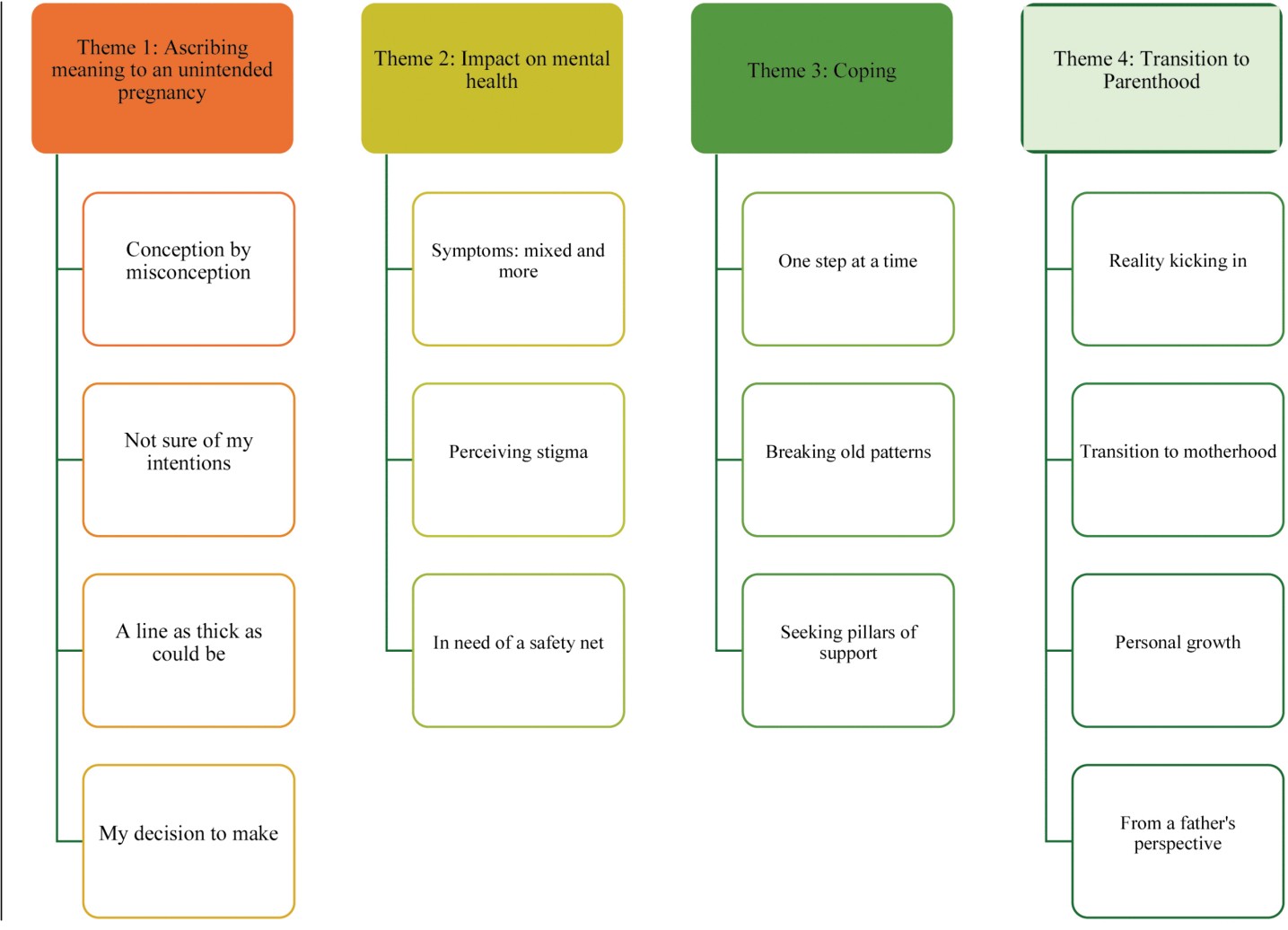

**Fig 1. Organization of themes and subthemes.** Legend: Legend: Four main themes with subthemes illustrating meaning-making, mental health impact, coping, and transition to parenthood.

*"I was happy at the beginning, or so, because um, well, I've been doubting motherhood my whole life. […] So, I was happy because I thought; it's happening to me. And there's no better opportunity, because then I don't have to decide myself and then it's just very good."* (Participant 3)

It may be interpreted that making the decision to become a mother was something participant 3 did not want to do her-self. Participant 7 echoed this experience by commenting that she did not want to make the decision to become a mother unless she was sure she would be a good mother. The UP had cleared the road for her on that account, as now she did not have to take on the responsibility of deciding. More participants described the pregnancy as something happening to them, instead of an act in which they were involved.

*"I think otherwise, maybe I would have thought that I always thought I have to be very sure that I'll be a good mother. And as long as I'm not very sure about that, I don't want it. And now. [it's already here...]"* (Participant 7).

For two women, the pregnancy was unwanted. For participant 8, the pregnancy felt as a personal punishment from God for not being careful enough. She described that she felt a need to take responsibility for it herself and bear the consequences by herself.

*"I had to just go through with it and that's you know, my cross to bear and it's my fault because I wasn't careful enough. And yeah, that just you know, God punishing me."* (Participant 8)

Her repeated use of the first person, 'I', may be interpreted as her feeling lonely in the decision-making process and her fully taking responsibility for the situation herself.

For participant 5, pregnancy was mostly perceived as a threat that caused her anxiety. She foresaw a future in which she would become a mother through adoption, as she feared newborns. She had never considered carrying a pregnancy herself. When asked how she had experienced her pregnancy thus far, she expressed her fear of the changing life plan and prospects of pregnancy.

*"Not good, ehh, because I was not planning to have a baby. I never wanted. And so, I was not feeling okay at all. Because I, you know, when you don't plan something, you have a plan. And then suddenly everything changed. So, it's a bit shocking. And the fact that I, I never wanted a kid, it's kind of really a bit, yeah, it drove me a bit anxious and, like, the not knowing what to do and what I want [...]. It has not been easy. It's kind of hard for me."* (Participant 5)

The words she used to verbalize her account of the situation seemed to undervalue the severity of her mental health symptoms, which became apparent during the interview.

**A line as thick as could be.** UPs evoked varying reactions from the participants at the time the pregnancy was discovered; none of them were initially positive. The words the participants chose when verbalizing their discovery illustrated how they assigned meaning to them and expressed their shock and surprise. Participant 5 realized she forgot to mention her pregnancy in the interviews' introduction, exemplifying the denial of her pregnancy.

Interviewer: *"Ehm, well and of course, we are interviewing you as well because you are pregnant."*

Participant 5: *"Yeah, yeah, ah yes, I'm pregnant. [Laughing] Oh, my gosh."*

The laughing may have implied that she was ashamed and in shock by how she forgot to mention the pregnancy in her introduction. In contrast to her verbal denial of pregnancy, she shared her bodily experiences. She explained that she could not sleep anymore, implying that there were things changing due to her pregnancy, but the pregnancy was not yet conscious.

*"[...] The only thing is that I'm not sleeping anymore. I'm having troubles, sleeping. But I don't have anything [pregnancy symptoms] right. I just feel, I still didn't digest the idea that I'm pregnant. I think when I talk, I'm like, okay, if you see in the introduction, I even forgot about it, right? It's kind of for me, it's. Yeah. It's, ehh, it has been hard to admit that I'm pregnant. That's the truth."* (Participant 5)

Another participant used various words, which gave a negative tone to the discovery.

*"And the line was like as thick as it could be, it was like 'you are pregnant'. I was just like 'God dammit'."* (Participant 4)

Participant 6 started speaking quickly and repeating her words when sharing the discovery of the pregnancy. Her words mark astonishment but also acceptance.

*"So ehm, so with my boyfriend, we actually met only like six weeks before we learned that we are pregnant. So that was like, we were like, oh my God. That's, that's way, that's way too soon, right? And it's like, and we, we didn't plan for it. We didn't expect it. It happened. It's, it happens."* (Participant 6)

**My decision to make.** The decision to continue or terminate the pregnancy was dependent on intrinsic and extrinsic factors. The most important intrinsic factor was the presence or absence of a (latent) desire for children and of a future family. The following participants responded that they would want to have children in the future: *"We knew we wanted to have children"* (participant 4) and *"We always wanted to, we knew we wanted children."* (Participant 2), *"And I was also very clear since the very beginning that I, I do want to have kids."* (Participant 6). Aside from a desire for a family, women's age and fear of not conceiving in the future played a part in accepting UP. Participants valued bringing a child into the world after having met certain conditions, such as having a paid job, housing and adult age.

*"Just getting everything a bit sorted indeed, and basically, you know, we both have good jobs, and we can financially raise a child just fine, we have the space for it, so yeah, if we both completely stand behind it, then there's actually nothing that, yeah, shouldn't go well now."* (Participant 1)

Seven participants considered being in a stable relationship to be an important condition for pregnancy; the possibility of becoming a single mother was negatively seen. The role of a supporting partner was specifically discussed by participants as an extrinsic factor that influenced the decision-making process.

*"[…] I would never want to do this alone. Like if I was unexpectedly pregnant and I wasn't in a relationship. Yeah. with someone who I would not feel confident having a child with, like I would have had an abortion, no question."* (Participant 4)

The decision to maintain or abort the pregnancy was mostly made with the partner. This was illustrated by participants 6 and 7, who repeatedly used the 'we' form when discussing the decision. They illustrate how the process of becoming pregnant was a shared responsibility and shared decision.

*"[…] So, it's 100% on us. It's like, […] we are adults, we are smart. We know what could have happened, right? So, it's like, so in this sense, it was no one's fault or like ehh, no one did anything by purpose or whatever. It's just, yeah, we just, we were just not careful enough."* (Participant 6)

This was at odds with the experience of participant 8, who felt pressured in her decision to maintain pregnancy. Her partner's wish was to maintain pregnancy as the dominant argument in her decision-making process.

*"My decision to keep it wasn't really my own. It was more, I think, fear, that if it ever did come out that would be something unforgivable to my husband."* (Participant 8)

However, in a different manner, participant 7 joins in the impact of the partner's perspective. She would not want to force the pregnancy on her partner and could only pursue the pregnancy with his support.

Participant 1 and her partner shared similar thoughts on the difficulty of decision-making. They pointed out how the woman is the one making the decision as she carries the pregnancy. This was a challenge for participant 1, as she wanted to maintain pregnancy, which was initially in contrast with her partner. For her partner, it was a challenge because he had doubts about continuing the pregnancy, but he agreed that she was in charge of the decision.

*"Yes, initially she was of course a bit sad that I sort of hinted in my response that I wasn't actually ready yet, or something like that. While of course, the choice actually always lies with the woman initially, right, when a woman is pregnant. So, um, yeah, that was a bit difficult, but well."* (Partner participant 1)

Participant 9 also doubted that she should continue her pregnancy, as the biological father was not involved. After consideration together with her general practitioner, she understood that her cognitive approach to decision making was not helping her. She shared that, resulting from her youth trauma, she had difficulty verbalizing her feelings and emotions. Thus, when she found herself buying a toy, it was clear for her she wanted to become a mother. This experience was a turning point for her. When talking about it, she cried, indicating the emotional significance of this turning point. In addition, she verbalized how this turning point developed her from an individual with a dilemma to a team (she and her baby) with a new purpose:

*"In the meantime, of course, there's a lot going on inside you. You start weighing all sorts of considerations and, um, yeah, then I bought a little cuddly toy. So that was actually, the sign for me, that I couldn't... [emotional]. […] And after all sorts of questions and answers, it was just clear: this is how it's going to be and, um. And we're just going to do it together. She and I together. Yes."* (Participant 9)

**Impact on mental health**

For all women, UPs impacted their mental health during pregnancy. UPs interfered with preexisting mental health symptoms or caused new-onset symptoms. The subthemes below describe what symptoms occurred, how they impacted women's journeys and how they fueled a desire for professional help.

**Symptoms: mixed and more.** A range of mental health symptoms, categorized into three groups, illustrated women's pregnancy journeys. The first group of symptoms was marked by childhood memories, as participants had negative parenting experiences during their own upbringing. For participant 2, her symptoms of posttraumatic stress disorder resurfaced during pregnancy. Her childhood flashbacks caused a burden during the first trimester of pregnancy.

*"And that actually made me very bleak, and yeah, I felt really bleak all the time. Because I also kept having those flashbacks all the time. […] and that, I think, made it so heavy."* (Participant 2)

Participant 9 had vivid dreams about her upbringing. She herself commented that the resurfacing of her youth trauma subconsciously occurred, as occurred in her dreams. She mentioned that she has difficulty verbalizing her feelings due to her childhood traumatic experiences, but in her dreams, the feelings appear.

*"I think that happens a bit subconsciously. Because I dream a lot about that [upbringing]."* (Participant 9)

The second group of symptoms is related to the unexpectedness of the pregnancy, which had a profound impact on participants' mental health, especially during the first and second trimesters of the pregnancy. For participant 8, mixed emotions marked her pregnancy journey. She also expressed anger toward her partner and toward herself for letting this pregnancy happen. This anger was fueled by the fact that she had a previous UP. The words used by the participants mark this emotion.

*"And at this point I'm blaming myself a lot more. You know like anybody can make a mistake once, but twice? Now I blame myself a lot. And I blame, you know, my husband."* (Participant 8)

Participants 3 and 4 described the emotions related to the UP as grief. These experiences were linked to mourning the past life as an individual (participant 3) and feeling loss of bodily autonomy (participant 4).

> "The fear is totally focused on my own life. So, I really had to, really mourn, I thought. I really felt grief coming over me." (Participant 3)

> "I still don't like being pregnant, I don't like it. I feel like my body is like, like I keep calling myself like 'the vessel'. I feel like my body is taking over not just physically but hormonally speaking and mentally. […] Like I miss feeling like me." (Participant 4)

For some participants, these emotions resulted in a prolonged state of anxiety or depression.

> "I just kind of feel resigned and defeated like… It's over, I give up. You know, stop having dreams. Because something will come along to dash it, so." (Participant 8)

Several participants described how the uncertainty of the situation led them to despair. Participant 1 felt like she was "going crazy" because pregnancy was something she absolutely did not want. Participants 3 and 5 felt so desperate that suicidal thoughts arose.

> "It was just a low point. I just wanted to get rid of it [pregnancy]. Besides being suicidal, I also just thought; get it out. I don't want this anymore." (Participant 3)

Finally, the third group of symptoms is inherent to the pregnancy itself, as pregnancy changed women's preexisting psychiatric symptoms and caused new symptoms, such as nausea, mood swings and irritability:

> "And uh, that mainly has to do with the increase in uh, compulsive traits in my personality. So uh, that became bothersome at a certain point, and uh, probably also a combination of hormonal factors along with, uh, a certain psychiatric sensitivity, I call it that now. Uh, that kind of triggered each other a bit more and that came to the forefront again." (Participant 9)

For participant 1, symptoms that she previously experienced due to her bipolar disorder resurfaced. Her partner, in his own interview, agreed with her by indicating how he felt about the pregnancy impacting her mood.

> "Um, she can occasionally react very sensitively and emotionally, and I think you see that during her pregnancy, influenced by hormones, a bit more strongly." (Partner participant 1)

A conversation between participant 2 and her partner (who were interviewed together) echoed this experience, where pregnant women and their partners had similar thoughts on the hormonal impact during pregnancy.
Partner: "That was really intense. It's tough that I... How? How can you turn this into a fight again?"
Participant 2: "Well, I didn't like it either because I was really caught off guard by that. Yeah, and that was all too much."
Although the emotional lives of women with UPs were marked by mixed and often negative emotions, UPs had several positive effects on them. For participant 4, the UP meant that she had successfully conceived, which caused her relief.

> "But I mean, I was talking like 'I'm healthy enough to get pregnant', and I can't believe like, in the back of my mind before I was like 'you know, I'll be sad if I can't have a child naturally'. I won't be devastated, but like I would feel sad. And I was worried like maybe I'm infertile or something. And like 'no you are not', so there was like a small silver lining there." (Participant 4)

 

**Perceiving stigma.** Most participants had doubts about sharing that the pregnancy was unintended with their social circle due to fear of negative reactions and stigma. Two participants shared how family members negatively reacted to their UPs.

*"I didn't tell my mom for a long time because she was very against the second unplanned pregnancy as well. […] It was, it was you know a lot of pressure on one side to keep it and be positive about it. And on the other side it was pure negative '[…], you've ruined your life', like, 'there's no way out of this'. So, not really support on either side."* (Participant 8)

Aside from not feeling any support, these reactions from significant others created hardship for the participants. Stigma was also related to mental health problems, introducing a double stigma: one of the UP and one of the mental health situations. This is illustrated by participant 3, who explains that she did not want to be open about her suicidal thoughts out of fear that child services would take her child away. She illustrated the fear of being perceived as a bad mother, which even made her suspicious.

*"At the beginning I was also afraid. There was a constant focus on 'suicidal, suicidal'. I thought; oh yes, if I now state very strongly that I'm suicidal, then my child will be taken away from me later, or something, it creates a kind of suspicion."* (Participant 3)

Participant 1 further elaborated on the stigma that she endured due to her mental health diagnosis. She feels that if something happens, others will attribute that to hearing a mental health diagnosis.

*"It's still something that isn't discussed indeed, and I'm just afraid, […], if they know that there's something, and something happens at some point, that it will immediately be attributed to that [diagnosis of bipolar disorder]. […] Yeah, so that, I'm not even going to touch that with a ten-foot pole."* (Participant 1)

Finally, some participants expressed the need for society to change its perspective on pregnancies and accommodate emotions other than happiness and content, as a pregnancy can be experienced very differently depending on the pregnant woman. Participant 3 experienced that positive reactions are the norm to any pregnancy, despite the actual feelings of the expectant parent(s).

*"Because then, yeah, I get a lot from many people: oh, congratulations! Oh, it's so fantastic! And then I reply: well, that doesn't apply to everyone. […] So, I do try to be more open about it, that not everyone is on cloud nine."* (Participant 3)

Participant 5 agreed and showed how she felt supported by mental health professionals (MHPs) who shared with her the stories of other women with psychiatric vulnerability during pregnancy. The normalization of negative (or neutral) reactions to a pregnancy made her feel better about the situation.

*"I think I created in my mind that every woman who gets pregnant is happy. And then the obstetric caregiver told me: well, it's not like this. […] There are a lot of women that suffer, that they have bad feelings, bad thoughts. And then I was like; okay, so at least I know that I am normal, let's say, right."* (Participant 5)

**In need of a safety net.** All participants were actively seeking mental health care and other forms of support with preventive measures to avoid worsening of symptoms during pregnancy and after childbirth. There were different

motivations to seek help: preventing relapse or transgenerational transmission, improving the baby's well-being, and preventing the worsening of psychiatric symptoms. Participant 1 found it particularly relevant to prevent postpartum relapse of her bipolar disorder in depression.

*"Because I do feel that I've done everything at least to prevent any depressions or issues."* (Participant 1)

Participant 7 stressed the importance of prevention; she tried to prevent difficulties with postpartum mother–child bonding by seeking help beforehand. Her transcript was marked by her wish to not perpetuate intergenerational patterns or traumas.

*"And seeking help or trying beforehand to do it as well as possible and seeking help in that process. I believe you have that responsibility as a parent."* (Participant 7)

Both participants 1 and 7 showed responsibility and an urge for prevention, not only treatment. Participant 9 prioritized her child's wellbeing by addressing it as '*the most important thing*' (participant 9). These words illustrated the significance of her child for her during pregnancy.

*"The most important thing for me was to get to the perinatal psychiatry outpatient clinic because there's also a pediatrician involved, […] the most important thing is that the unborn child isn't harmed."* (Participant 9)

Participant 3 echoed the view that taking care of your mental health situation is more crucial when a child is involved and has fueled her wish for help. Participant 5 also asked for treatment when she realized that her symptoms worsened.

*"I went to my doctor and then I said: I don't want to have a baby. I am going to get crazy. I'm going to kill myself because I don't want this. This cannot be happening. So, I was completely out of my mind and then I said: please, can you please help me with some... ehm, I need that mental health support."* (Participant 5)

Although the participants expressed their own motivation to seek help, there were concerns regarding the availability of mental health care for women with perinatal mental health problems. Participant 2 expressed that she lacked information about the effect of pregnancy on her psychiatric vulnerability and where she could find treatment, even though she had previously received psychiatric care. She shared that it would be more valuable if her MHP had informed her of the peripartum challenges prior to pregnancy.

*"But I think if there would be more information [...] whether a psychologist or so could offer something like that, if you're already seeing a psychologist, that might be an even better step."* (Participant 2)

Participant 3 hoped that healthcare professionals (HCPs) would pay more attention to mental health symptoms to ensure early treatment. She saw this opportunity for midwives.

*"So, in addition to the physical questions, maybe you could ask more, like: oh yeah, and where are your concerns? Do you have a lot of worries? How is it going inside your brain? Yeah, do you wake up feeling okay in the morning?"* (Participant 3)

## Coping

The third theme exemplifies how women coped with UP and co-occurring mental health symptoms.

**One step at a time.** Women adopted different coping mechanisms to address the UP. Participant 5 illustrated how she concentrated on each day, without looking at the future. This prevented her from feeling overwhelmed. At the start of the interview, she denied her pregnancy. This coping mechanism of not thinking about the future could indicate another form of denial of the current stressful situation that is the result of UPs.

*"I never wanted a kid, it's kind of really a bit, yeah, it drove me a bit anxious and, like, the not knowing what to do and what I want and ehm, but then. Well, I'm trying to, to concentrate on each day."* (Participant 5)

Participant 8 described how distraction was the way for her to feel better.

*"I don't always want to sit there just like talk about my feelings because my feelings are very negative. So to feel positive, I need to look outward of myself"* (Participant 8)

Participant 6 commented on another helpful aspect of any pregnancy: its duration. The passage of time helped her in accepting the pregnancy and adjusting to the new reality.

*"I was super, super weak, he [her partner] was depressed and it was like, completely, you know, ups and downs all the time. So that was not very pleasant, to be honest. But that passed for both of us. And more or less since the second trimester started, it's like I started to feel better. He also started to feel better emotionally. And right now, we are looking forward to everything."* (Participant 6)

**Breaking old patterns.** Pregnancies made participants reflect on their own childhood and their parents' parenting skills. As all participants and one partner mentioned an unsafe upbringing, they were motivated to meet their babies with unconditional love, safety and healthy parenting skills and to break the chain of intergenerational trauma. Participant 1 comments on doing so differently.

*"We both have this feeling that we just don't want to do it the way our own parents did, actually."* (Participant 1)

For participant 2, the most difficult part was that she never experienced unconditional safety. She was aware of that and would love to provide it for her child. It frightened her that she has not had an example of this.

*"I don't know how it feels to have, sort of unconditional safety, um, because I've never experienced that, and I know; I'll never feel that. Um, and in that aspect, that's something I would really want, so I'm working very hard on that. At the same time, it's very frightening that you don't know how that feels. So, planning to give something you don't know."* (Participant 2)

Although women had often previously received professional help for psychiatric symptoms, the current pregnancy acted as the instigator of help-seeking behavior.

*"… Like I said earlier: I don't feel like I would be doing it for myself, it's really for, for my child. Yes, definitely. Because for myself I think; oh well, I'll survive, I've known that for 38 years, you know. Um, yeah, for your child, you just don't want that after childbirth, yeah, that seems truly terrible, the bonding and if that doesn't happen, um, yeah, that."* (Participant 3)

**Seeking pillars of support.** As illustrated in Theme 2, participants realized that they needed professional help for the mental health impact of UPs and that as a coping mechanism; they actively sought pillars of support. MHPs formed one

pillar of support, which positively affected the pregnancy process. Participants 3 and 4 both described how the help of an MHP is of additional value during their pregnancy, next to the help from family, friends and midwives.

*"I feel heard, which is really great. Besides having your family, it's just really nice to have professional people who, um, think along with you and are willing to do everything they can to help you, and that's special. I feel very supported in that, especially from the midwife who took action so quickly. My doctor was also very prompt."* (Participant 3)

Women also received support from other pillars of support such as friends and family.

*"[…] well, my father found it very nice, he was very enthusiastic when I called that I, uh, so that's nice. All my girlfriends were, uh, uh, quite supportive and they all found it, well, I also had a friend, you know, who was there when I did my pregnancy test."* (Participant 7)

Participant 1 shared how her partner's mother had a role in the decision-making process regarding the pregnancy. Her support helped her partner accept UP.

*"And at one point, I said to him: well, call your mother because I know he always listens to her, that he, well, can just have a good conversation with her. And after that phone call, he suddenly completely changed his mind, [...] then suddenly he was like, well, 'then we'll just do this', and um, then I also thought; 'huh, so yeah, that was a bit strange', but it did help in that sense indeed to really talk to people about it […]."* (Participant 1)

### Transition to Parenthood

The fourth theme illustrates how parenting began during pregnancy, as women had bonding experiences with their unborn child. The theme further describes how participants experienced the first weeks to months of parenthood compared to their expectations during pregnancy. Looking back on their pregnancy journeys, participants experienced growth on an intrapersonal and interpersonal level.

**Reality kicking in.** The bodily sensations that were related to pregnancy had different meanings. Participants felt as if the babies were 'real' when they experienced their movements. Feeling the baby also reassured the expectant mothers of the baby's health.

*"And when I started feeling it more, then it becomes… real, you know. This is real, there is really something in your belly. And as you're getting closer to the end, I do feel it more to a greater extent. Uh, throughout the whole day, it's more, well, more realistic or something like that."* (Participant 2)

Another meaning of fetal movements was the creation of a bond between expectant mothers and babies. Participant 4 described how she feels love toward her baby when she makes contact with her belly.

*"And I feel tenderness towards him, I feel a lot of love towards him. Like last night, normally he is very active at night, he wasn't active last night, and I was like I can't fall asleep until I felt him move. Because I, I was like I need to make sure he's still alive."* (Participant 4)

For participant 6, fetal movements had metaphorical significance. For her, the movements of her baby created imaginations of her future.

*"Yeah, I'm also like laughing a bit like that because she's moving really a lot. So, I'm laughing a bit that she will be either a dancer or a kickboxer in the future."* (Participant 6)

The reality of having a baby was also instilled by ultrasounds, which play an important role in visualizing the baby. This happened both literally and metaphorically. Participant 1 explained how she imagined her baby physically look like his father on ultrasound. Interestingly, she also comments on his character. As the baby is in a breech position, doing things differently from others. In this way, ultrasounds enhance fantasizing about the baby.

*"Yeah, they really saw those long legs in one of the first ultrasounds, I thought; oh, that's really my boyfriend, and in the 3D ultrasound, you can really see his face in there, so, yeah, I'm curious. So far, he's doing everything in his own way, the fact that he's even being born is, of course, not initially a choice from us. And, yeah, I just think he's very stubborn, he was in breech position, so, yeah."* (Participant 1)

Contrary to the positive feelings and ideations that resulted from fetal movements or visualizations of the baby on ultrasounds, fears regarding the pregnancy and the unborn child were also metaphorically described. Women gave substance to these fears by fantasizing about the baby.

*"I think once I know that, that I can begin to kind of, again like personify or humanize it a little bit more. Because at this point it's just a thing, like this unmortise thing that is in me and wreaking havoc on my sanity."* (Participant 8)

**Transition to motherhood.** As new parents, participants were surprised by the way in which bonding, love and protecting the baby could occur, even though not all participants had a healthy and safe parental example themselves. From the moment of birth, participants felt a shift in priorities. Love for the baby also prioritized their wellbeing over other aspects of life. Participant 2 illustrated this as being less willing to head back to work after her maternity leave.

*"I really feel such a bonding. […] Then there's really this kind of protective feeling that you couldn't even imagine during pregnancy. Because I really couldn't imagine that it's really there now. Mmm, that's very special to experience and very different from what I thought."* (Participant 2)

Participant 6 expressed her love for her baby through communication during the interview, which was marked by her talking to her baby, addressing her with positive nicknames and making nonverbal contact.

*"[talks to baby] Oh, yeah. Pure joy [name baby]. Pure joy. Yeah. All right. All right. Baa! Baa! Baa! Baa! Are you happy, baby? Now, Are you happy, baby? Now? Yes."* (Participant 6)

Participant 1 showed how she prioritized her baby by using superlatives.

*"Yeah, he's truly the most important thing in my life, and I'd do anything for him. The most important thing for me is that everything goes well with him and that he becomes happy."* (Participant 1)

Prioritizing the baby's wellbeing also had its downside. For participant 7, her focus on preventing trauma to her baby put considerable pressure on her. She experienced the responsibility of parenting as a challenge.

Participant 7: *"I find the responsibility sometimes a bit overwhelming, you know. Suddenly, with everything, you think, 'Okay, but what if I mess this up?' And I know it doesn't happen because of one thing, but sometimes I do think that."*

Interviewer: *"And what kind of things are those?"*

Participant 7: *"Well, first they said you should look at him all the time when you breastfeed him, and I just can't do that. […] I do find it occasionally quite difficult when I think, 'Oh, but what if he looks at me and I don't look back? Will it ruin him?'"*

**Personal growth.** Looking back on their UPs, personal growth was noted in terms of declining mental health symptoms, finding a purpose in life, experiencing love in new ways and finding better ways to communicate within (partner) relationships. Participant 2 explained that she felt an urge to continue with her life for her child, even though negative emotions were still part of her daily life.

*"Also, that even if I am anxious or something, I still have to carry on. […] That I have to be there for her or something."* (Participant 2)

Looking back on her pregnancy experiences encompasses her fear at the very beginning of the pregnancy journey.

*"Yes, so about getting unintentionally pregnant indeed. And then I found it very scary and thought it's not the right moment. But now I really feel like I would never have wanted it any other way. I actually could have had children earlier because my body just knew it so naturally. We both feel like it has truly enriched us."* (Participant 2)

Participant 4 expressed surprise with the way things turned out by saying '*it's funny*'. She never expected this UP to turn into this amount of love for her child. By calling it '*magic*,' she marks the significance of this experience.

*"I mean, it's, it's funny like, yeah, he was unplanned, but, like, I'm so happy that he's here. […] part of my heart is, like, now exists here and, and like, I love him so much. Like I never knew I could love this much and, like, feel this like, joy. I never felt anything like this before. Like, I really feel like I've, like, unlocked a new world of, like, internal joy and love and, like, magic that I never thought was possible."* (Participant 4)

Participant 7 found a positive aspect of the growth of her relationship with her partner. Her child makes her feel a need to repair after conflict, whereas in the past, she wanted to leave in conflict.

*"I think my relationship with [partner] has improved in that aspect. But more so, I think I've become better at it. In the sense that in the past, I had more of the feeling that I could always leave."* (Participant 7)

For participant 1, the UP motivated her to set boundaries. Before her pregnancy, this was challenging for her. During pregnancy, her motivation was fueled by the protective feelings of the unborn child.

*"I'm especially concerned that I don't want my child to deal with those kinds of things, with that irrationality and that he'll have to endure such things in his life. That's why I don't have any contact with my sister anymore, I've completely cut her off. I just don't want those kinds of people in his life."* (Participant 1)

**From a father's perspective.** Participant 2 and her partner emphasized the positive impact of pregnancy on her mental health by providing purpose to her life.

Participant 2: *"Yes, oh yeah, that's something too. Yeah, that's... Maybe a very good reason. I also notice that I feel much better than even before the pregnancy because now I can't just lie in bed forever or something like that. [...] and I can still be scared, but for a shorter period or something. Mmm. So, I also feel much less lonely or something."*

Partner: *"That's funny. We've talked about whether we should still, because I didn't want a pet at that point. Or whether we should get a cat or something because that can actually have quite a positive influence. [...] But we always knew*

*that. That something like that [a baby]. Because even when we looked after a dog for a week or so and you were just really, even though that wasn't actually such a good time for you, you felt uh much better when you were taking care of something."*

The perspective of the partner of participant 1 on her pregnancy journey emphasizes that UP can lead to personal growth for women with psychiatric vulnerability.

*"Look, it's very good that there's obviously monitoring and follow-up for patients like [name participant 1], but I really think she's making a tremendous effort and that she's perfectly stable. […] What I'm actually trying to say is that it's good that those people [with psychiatric vulnerability] are being followed, you know, and monitored and that there is follow-up. But I think credit should also really go to those people who seem to be functioning better because of such a pregnancy. So, I'm happy about that and also proud."* (Partner of participant 1)

## Discussion

This study aimed to understand how women with psychiatric vulnerability experience UPs. Four major themes were derived from the interviews with nine pregnant women and two partners.

The 'Ascribing meaning to an unintended pregnancy' theme described how UPs could occur in the context of women's situations. Participants' pregnancy intentions prior to pregnancy were often ambivalent and while some women actively tried to avoid pregnancy, others did not. In a previous study, pregnancy ambivalence is an independent risk factor for UPs [21]. Moreover, a previous study demonstrated how incongruence between pregnancy desires and planning can cause internal or external conflicts, such as desiring a pregnancy but also experiencing financial worries [21]. In our sample, fear of motherhood concurrent with the desire to create a family might explain the incongruence between pregnancy desires and pregnancy planning. Fear of transgenerational transmission of trauma and psychiatric symptomatology impacted women's desire for a future family and instilled fear of inadequacy in parenting. Another factor that contributed to some participants' UPs was misconception regarding their fertility; as was found in the study by Borrero et al [44]. Finally, in contrast to the previous literature, our participants did not show that mental health symptoms such as lack of overview, depressed mood or hypersexuality made them susceptible to UPs [42,45].

The 'Impact on mental health' theme illustrated how UPs caused mental health symptoms related to childhood memories, the unexpectedness of the pregnancy and the pregnancy itself. Moreover, women had difficulty announcing their UP due to stigmas. Mental health symptoms during pregnancy, in addition to a fear of transmitting childhood traumatic experiences and being an inadequate parent, caused despair. Their responses, summarized by some women as 'grief coming over me', resemble the five stages of grief that can be identified in persons who are mourning: denial, anger, bargaining, sadness, and acceptance [46]. Parents without psychiatric vulnerability also reported a profound emotional impact resulting from UPs, including mental health conditions, postpartum depression, suicide attempts or hospitalization [47]. However, the notion of dreams about childhood, flashbacks to their childhood trauma and fear of transgenerational transmission of trauma in our sample has not been previously described in relation to UPs. It is widely known that maternal representations, often present even before pregnancy, are derived from women's own upbringing and parental experiences [48]. Maternal representations of pregnancies and babies as scary, monstrous or negative, as in our sample, could be related to these past experiences. These representations might impact future mother–infant attachment [48]. As it is possible to improve attachment problems between mothers and their offspring [49], it is valuable to understand women's maternal representations and their origin. This might be even more relevant for women with childhood traumatic experiences.

In the 'Coping' theme, coping mechanisms such as focusing on the positive or on the future, seeking distraction and allowing support were discussed. The 'Transition to Parenthood' theme linked imaginations and expectations during

pregnancy with postpartum experiences. Ultrasounds and feeling the baby kick were important milestones in pregnancy acceptance as well as potential bonding experiences for pregnant women and their partners. These milestones consolidated the pregnancy and provided reassurance about the health of the baby. In interviews with women without psychiatric vulnerability, similar positive effects of ultrasounds were found in the first trimester [50]. These experiences could be points of reference in supporting women with UPs. In six postpartum narratives, awareness of psychiatric vulnerabilities gave rise to behavioral changes, such as creating safety nets, setting boundaries and prioritizing their babies. For some women, mental health symptoms improved as the UP provided a new purpose in life. Two partners underscored this perspective. Although UPs can be perceived as challenging pregnancies, our participants showed how they could also provide a window of opportunity for personal growth. Previous qualitative research shows how parents are motivated by UPs to make lifestyle changes, as UPs give a renewed sense of purpose [47]. The most compelling finding in our study is that in women with psychiatric vulnerability, UPs can install a window of opportunity for personal growth by changing behaviors or receiving treatment as participants realize how their psychiatric vulnerability and/or adverse childhood experiences might impact future parenting. A quantitative study with couples who had UPs also reported improved relationship functioning after birth compared with before birth [51].

In an Iranian interview study on women with UPs conducted in a different social and cultural setting, feelings of shame, guilt and self-blame were also reported, suggesting that societal stigma may impact pregnancy journeys in various social settings [23]. Although pregnant women in our study experienced support from their partners, MHPs, friends and/or family, two women feared becoming a single mother. They emphasized that continuing the pregnancy was dependent on partner support, which is in accordance with the findings of a recent study that highlighted the importance of partner support for pregnant women's psychological wellbeing [52]. This finding may also reflect a lack of support and accommodation provided by contemporary society for single parents. In our study, the future fathers' feelings toward the pregnancy ultimately aligned with those of the women. Additionally, they perceived the pregnancy as positive for the mental health and personal development of their partner. Previous studies have reported conflicting evidence on how pregnant women and their partners may or may not run parallel in their feelings about continuing or terminating their pregnancy [31,53]. One study found great heterogeneity in how their role in the decision-making process was perceived [54].

This study has several strengths. To date, no other study has explored the unique experiences of UPs in women with psychiatric vulnerability using qualitative data. The interview guide was developed following focus groups with experts with lived experience with UPs and psychiatric vulnerability, adding suitability and appropriateness. The small sample size is regarded as a strength, as it serves the IPA in prioritizing individual experiences [34]. The included women were homogenous in the presence of a psychiatric disorder, enhancing similarity, however we included women with various psychiatric disorders, which provided a deeper understanding from a transdiagnostic viewpoint [45]. We invited involved partners as partner experiences are understudied in the context of UPs. However, the interviews with the partners reflected their perspectives on their spouses' experiences rather than their own. Ad-verbatim transcribed interviews were analyzed for both semantic content and form: emotional responses, metaphors and nonverbal communication were used to attribute meaning to the transcripts. We further adhered to the phenomenological standpoint by including bodily experiences in the interpretation of findings such as feeling kicks of the baby [39]. By adhering to the seven-step analytical plan of Charlick et al., we increased procedural precision [33].

This study has several limitations. While participants were offered to review the transcription and summary of their interview, they were not involved in the analysis and writing of the results. In our sample, all participants were employed and living independently, indicating a middle to high socioeconomic status, that may have influenced their ability to continue the pregnancy for financial reasons. Beyond financial resources, higher socioeconomic status is associated with broader structural advantages, including better access to healthcare, stable housing, health literacy, and supportive social networks. As articulated in Ross's framework of reproductive justice, access to healthcare and housing, as well as the presence of a support system, are fundamental factors (and basic human rights) when considering whether to continue

an unintended pregnancy [55]. In samples with fewer socioeconomic resources, financial insecurity and related structural barriers may therefore play a more prominent role in shaping pregnancy-related decision-making. All but one of the women were pregnant with their first child. This may indicate selection bias, as women with (multiple) children may not have time to participate in an interview. One of the inclusion criteria was that participants speak Dutch or English fluently, leaving out people who might have different needs and experiences due to language or cultural barriers or discrimination in healthcare. This reflects the investigators' demographics, as the study group consisted of native Dutch women proficient in Dutch and English. This may have caused a threshold for women from other demographic groups to join the study. The outpatient clinic is an advisory clinic, which indicates that all participants voluntarily consulted on their mental health, implying a certain degree of mental health awareness. Because some interviews were conducted with partners present while others were conducted individually, the presence or absence of support persons may have influenced participants' willingness to disclose sensitive emotional experiences or relationship dynamics, potentially shaping the depth and direction of the narratives. In addition, the interview guide was iteratively adapted throughout data collection. While this flexibility supported participant-led storytelling and allowed emerging issues to be explored, it may also have affected participants' responsiveness to specific prompts and introduced variability in the salience and depth of topics across interviews, thereby influencing the nature of the data obtained. Furthermore, there is a possibility of response bias, but we consider this bias limited because the participants' responses were diverse and captured a range of emotions and reactions. Several issues are related to the data collection. Diversity in the timing of the interview might have increased recall bias. Finally, while we made use of triangulation, we did not use bracketing to reduce authors' biases. Pregnancy intentions are difficult to recollect after a pregnancy is accepted, especially as mothers' bonding can increase during pregnancy and change recollections [56]. Postpartum interviews were held with only six participants, which undermines the prospective character of this study. Participants who did not join might have experienced pregnancy loss, complicated pregnancies or postpartum mental health problems. The postpartum perspectives may thus be marked by those with positive pregnancy and birth experiences. However, the experience of UPs and the transition to parenthood can be best studied with a prospective approach [57]. We therefore decided to incorporate the perspectives of the remaining participants. Finally, as all pregnant women and partners discovered UP in the first trimester of pregnancy, there was a possibility for legal abortion. This is relevant, since having the right to have a choice regarding maintaining the pregnancy or not has important emotional, social and economic consequences [57].

To date, many studies have focused on the adverse outcomes of pregnancies in women with psychiatric vulnerabilities, and potentially positive effects may have been overlooked [11–13,58]. Future studies may investigate other effects of UPs such as the possibilities for relational growth, personal growth and seeking mental health support. A deeper understanding of women's needs could help policy makers develop tailored interventions to improve pregnancy decision making and, if desired, pregnancy acceptance and maternal-infant bonding [49,58,59].

The current clinical interventions available for this subgroup of women with psychiatric vulnerability in relation to UPs are limited. A scoping review on the transition to motherhood previously issued a warning that it would be a waste not to utilize the potential for growth during the transition to motherhood [60]. Based on our findings, HCPs should routinely screen for pregnancy intention in women with psychiatric vulnerabilities and offer counselling to explore possible ambivalences and consider the social conditions and possible previous traumatic experiences. Furthermore, they should encourage the involvement of partners and other pillars of support, and discuss possible emotions regarding bodily sensations of the baby or visualization thereof via the ultrasound. Initially unplanned but finally welcomed pregnancies could provide a window of opportunity to obtain appropriate psychiatric healthcare and tackle transgenerational patterns.

## Conclusions

This research reveals novel perspectives on how women with psychiatric vulnerability experience unintended pregnancies. Through a phenomenological lens, we delineated how UPs impacted pregnancy journeys in our sample and

triggered reflections on childhood experiences. UP journeys differed among women with psychiatric vulnerability, and the decision to carry the pregnancy was made in a personal way. Despite the perception of UPs as challenging pregnancies, in our sample UPs may help to create a momentum to engage in mental health treatment and might offer the potential for personal growth. The narratives from our work provide further understanding of women's experiences among MHPs and may help them to support comprehensively expectant parents with psychiatric vulnerabilities. Finally, society challenges pregnant women with psychiatric vulnerability and UPs by imposing stigma and expectations. MHPs could help decrease the pressure and stigma that impacts UPs by openly and promptly discussing UPs without making judgments about their patients and listening to their needs.

## Supporting information

**S1 Appendix. Antepartum interview.**
(DOCX)

## Acknowledgments

We express our gratitude to all the participants who were willing to share their stories in this study. Additionally, we are appreciative of Hanna Salverda for her contribution to the interviews. We thank H. Heller for her role as an independent consulting physician for the participants and M. Rexwinkel for her contribution to the interview guide.

## Author contributions

**Conceptualization:** Noralie Schonewille, Elena Soldati, Monique van den Eijnden, Maria van Pampus, Thomas Zoon, Odile van den Heuvel, Birit Broekman.

**Data curation:** Noralie Schonewille.

**Formal analysis:** Noralie Schonewille, Elena Soldati.

**Funding acquisition:** Noralie Schonewille, Monique van den Eijnden, Nini Jonkman, Maria van Pampus, Odile van den Heuvel, Birit Broekman.

**Investigation:** Noralie Schonewille, Monique van den Eijnden.

**Methodology:** Noralie Schonewille, Elena Soldati, Monique van den Eijnden, Nini Jonkman, Maria van Pampus, Odile van den Heuvel, Birit Broekman.

**Project administration:** Noralie Schonewille.

**Resources:** Noralie Schonewille.

**Supervision:** Birit Broekman.

**Writing – original draft:** Noralie Schonewille, Elena Soldati.

**Writing – review & editing:** Monique van den Eijnden, Nini Jonkman, Maria van Pampus, Thomas Zoon, Odile van den Heuvel, Birit Broekman.

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
