## [Decision Letter · Decision Letter 0]

17 Nov 2025

Dear Dr. Soldati,

Thank you for submitting your manuscript to PLOS ONE. After careful consideration, we feel that it has merit but does not fully meet PLOS ONE’s publication criteria as it currently stands. Therefore, we invite you to submit a revised version of the manuscript that addresses the points raised during the review process.

We look forward to receiving your revised manuscript.

Kind regards,

Taiwo Opeyemi Aremu, MD, MPH, PhD

Academic Editor

PLOS ONE

[This research was funded by ZonMw, grant number 554002007].

3. In the online submission form, you indicated that [The data that support the findings of this study are available from the corresponding author upon reasonable request.].

Additional Editor Comments (if provided):

Reviewers' comments:

Reviewer's Responses to Questions

**Comments to the Author**

1. Is the manuscript technically sound, and do the data support the conclusions?

Reviewer #1: Partly

Reviewer #2: Partly

2. Has the statistical analysis been performed appropriately and rigorously?

Reviewer #1: N/A

Reviewer #2: N/A

3. Have the authors made all data underlying the findings in their manuscript fully available?

Reviewer #1: Yes

Reviewer #2: No

4. Is the manuscript presented in an intelligible fashion and written in standard English?

Reviewer #1: Yes

Reviewer #2: Yes

Reviewer #1: Abstract

In the results section, rather than describing the themes, I recommend summarizing the key findings or main takeaways to provide a clear overview.

Background

Line 93-97 should be rephrased to improve clarity

The background section would benefit from additional depth, including evidence from previous studies and a more detailed explanation of the challenges faced by women with UPs.

The study objectives (line 103-109) should be stated more clearly

Methods

The manuscript appear to follow the JARS-Qual checklist instead of the COREQ or SRQR checklist, which are required by PLOS ONE. Please ensure compliance with journal guidelines.

Clarify what information was included in the patient files (line146) and how this data was analyzed.

Consider providing the interview guide as a supplementary file

Explain the process for translating the interviews conducted in Dutch into English

Provide details on how demographic data were analyzed.

Results

In the demographics section:

- Include the attrition rate for the second round of interviews

- Add mean age and key participant characteristics in the text.

Line 193-198 – recommend removing this section

Line 262-263 – Recommend removing this section

Theme 1. While the results provide useful context, consider making this section more concise.

Discussion

The “Main Findings” section should be condensed and integrated with “Interpretation of findings”. I suggest removing the subheading and presenting the content under a single “Discussion” heading.

Include practical recommendations to strengthen the discussion.

Move the “Strengths & Limitations” section immediately before the conclusion and make it more concise.

Combine the “Implications for Future Research” with “Strengths & Limitations”, and shorten this combined section.

Reviewer #2: Introduction: The introduction introduces the reader to the state of the literature about unintended pregnancies and provides a global statistic which highlights the significance of the topic. In line 88-90, the adverse effects of psychiatric vulnerability were described, however, the authors do not describe the mechanism by which these psychiatric vulnerabilities contribute to increased risk of UPs and among which populations (if there are existing disparities). In line 94, review language used when attributing UP to women experiencing psychiatric vulnerability. For example, "...UP among women..." would be more appropriate as the term emphasizes the collective. In line 103-104, there appears to be a misalignment between the gap stated in line 102; ongoing pregnancies is a new term that was not discussed elsewhere in the introduction nor was it defined. To strengthen this section, describe the state of the literature on ongoing pregnancies one this term has been defined as you have done with UP. The gap you have identified should reflect that to date, there has not been a study that explores how women with psychiatric vulnerability experience UP and ongoing pregnancy. In line 104, the objectives are unclear. The term "besides" is informal. Please review the manuscript for similar instances. It appears that your objectives are to (1) identify and describe how UP occurs among women with psychiatric vulnerability and (2) factors that influence decisions around maintaining the pregnancy. In line 107-109, you have introduced a new gap in the literature. The first time it is mentioned should be earlier in the background, not as part of the aims. In doing so, justification for the objectives is substantiated. All persons may not be familiar with family planning which was mentioned twice in the introduction. It may beneficial to define it in this context.

Methods:

Research Study Design: In lines 115-117, there are no citations to support the description of prospective qualitative analysis. To avoid unintentional plagiarism, ensure all ideas that are not originally yours are cited. In 118, a narrative approach was used, however, you fail to describe the appropriate of understanding the construction and meaning of the experience. This is different than phenomenological research from which you would extract meaning from shared experiences among these women to understand the essence of UP. Justification on the narrative approach for this population is warranted. It is unclear how this approach was used and how phenomenological underpinnings were incorporated in extracting meaning from these stories.

Researcher Description: The researchers have thoroughly described their positions but did not disclose the intersection of other identities that may impact their knowledge of the phenomenon, responsiveness, and interpretation of the results. Positionality was alluded to but there is no reference to power dynamics that may have existed based on these identities, both with the study participants and amongst the researchers. There is no description about reflexivity and methods used to address psychological and emotional responses to the data, bracketing, consultation, or other means for example to reduce bias. Failing to provide this information significantly impacts rigor and transparency. The researcher is also an instrument in qualitative inquiry.

Participant Selection and Recruitment: In line 139-140, the type of recruitment (e.g., snowball, convenience, purposive, etc.) needs to be specified. Review this segment as there are some missing punctuations and issues with grammar. For example, in line 147, the sentence should read, "Participants were asked..." instead of "was asked." In line 147-149, clarity is needed on criteria for inviting persons to interviews and partners participating in interviews. Asking participants if their partners wished to participate is called snowball sampling which means you have used both purposive for your primary participants and snowball sampling for recruiting their partners. There is no description about parters receiving informed consent separate from their partner.

Data Collection: In line 159-160, describe your approach to maintaining confidentiality for persons interviewing at the hospital. For in-person interviews, what software or device was used to record the interviews?

Data Analysis: In line 164, specify if a third-party transcribed the interviews OR if the researchers transcribed the interviews independently. Did researchers review the transcriptions and reconcile discrepancies? This information is crucial in ensuring the participants experiences were captured.

Results: In line 187, there are noticeable differences in when follow-up interviews were conducted. In reviewing your methods section, the follow-up time for the second interview was not specified. State the intend follow-up time for the second interview (e.g., estimated 8-weeks postpartum) and if this was not accomplished, factors that influenced follow-ups. The remainder of the results were comprehensive, with select quotes accurately reflecting themes and sub-themes that emerged in the data. Quotes captured the essence of patients' experiences. The researchers did not attempt to interpret these findings and maintained a neutral position by articulating what participants' stated.

Discussion: In lines in 659-681, the discussion section is not designed to restate what is in the results. This information should be interwoven into the interpretation section.

Strengths & Limitations: In line 701-702, there appears to be a typo. Clarify if you are referring to women of low- to middle-class or SES and how this can influence decisions in family planning. There is no mention of instrument limitations for other methodological limitations. For example, how did modifying the semi-structured interview possibly impact responsiveness to prompts and data that was obtained? Examine the limitations of narrative and phenomenological inquiry and the sensitivity of this topic. Furthermore, there is no description of how the presence or absence of partners and support persons may have influenced responses during the interviews.

General: Ensure the interview guide is available in an appendix.

**Do you want your identity to be public for this peer review?** For information about this choice, including consent withdrawal, please see our Privacy Policy

Reviewer #1: No

Reviewer #2: **Yes:** Shane´ J. Gill

---

## [Author Response · Author response to Decision Letter 1]

19 Jan 2026

Thankyou for your time and vaulable feedback

---

## [Decision Letter · Decision Letter 1]

2 Feb 2026

Exploring unintended pregnancy journeys among women with psychiatric vulnerability using interpretative phenomenological analysis

PONE-D-25-33332R1

Dear Dr. Soldati,

We’re pleased to inform you that your manuscript has been judged scientifically suitable for publication and will be formally accepted for publication once it meets all outstanding technical requirements.

Kind regards,

Taiwo Opeyemi Aremu, MD, MPH, PhD

Academic Editor

PLOS One

Additional Editor Comments (optional):

Reviewers' comments:

Reviewer's Responses to Questions

**Comments to the Author**

Reviewer #2: All comments have been addressed

2. Is the manuscript technically sound, and do the data support the conclusions?

Reviewer #2: Yes

3. Has the statistical analysis been performed appropriately and rigorously?

Reviewer #2: N/A

4. Have the authors made all data underlying the findings in their manuscript fully available?

Reviewer #2: Yes

5. Is the manuscript presented in an intelligible fashion and written in standard English?

Reviewer #2: Yes

Reviewer #2: The authors have thoroughly addressed the recommendations provided in the initial review. In the introduction, the authors introduced substantive literature which was sufficiently synthesized, to provide a comparison of factors that influence psychiatric outcomes among women that experience unintended outcomes. Due diligence was taken to identify differences that may exist within this population, with unique vulnerabilities that may vary by the type of psychiatric diagnosis. In doing so, the authors indirectly address health equity which is central to the literature and manuscript. The aims of the study were described and are consistent with the gap in the literature that the authors identified.

Positionality and reflexivity were added to the manuscript with a robust description of each researchers' respective identities, intersectionality, and the meaning-making, all of which increased rigor and transparency. Furthermore, the availability of this information improves opportunities for replication. To this end, clarification of sampling was provided and aligned with the design and method. The researchers inclusion of a statement of anonymity in data analysis also supports rigor and reduction of bias which are essential in qualitative research. The discussion is more robust compared to the first submission, with repetitive information removed and thorough synthesis of earlier research, allowing for comparison. I thoroughly appreciated the fullness of the limitations section which included a framework in explaining how a reduced sample size may with structural and systemic limitations may shape anticipated effects.

**Do you want your identity to be public for this peer review?** For information about this choice, including consent withdrawal, please see our Privacy Policy

Reviewer #2: **Yes:** Shané Janelle Gill

---

## [Editor Report · Acceptance letter]

PONE-D-25-33332R1

PLOS One

Dear Dr. Soldati,

I'm pleased to inform you that your manuscript has been deemed suitable for publication in PLOS One. Congratulations! Your manuscript is now being handed over to our production team.

Kind regards,

on behalf of

Dr. Taiwo Opeyemi Aremu

Academic Editor

PLOS One